# Insect Cultural Services: How Insects Have Changed Our Lives and How Can We Do Better for Them

**DOI:** 10.3390/insects12050377

**Published:** 2021-04-22

**Authors:** Natalie E. Duffus, Craig R. Christie, Juliano Morimoto

**Affiliations:** School of Biological Sciences, University of Aberdeen, Zoology Building, Tillydrone Ave, Aberdeen AB24 2TZ, UK; natalie.duffus.17@abdn.ac.uk (N.E.D.); craig.christie@abdn.ac.uk (C.R.C.)

**Keywords:** sustainability, Sustainable Development Goals, biodiversity, policy, ecosystem services, societies

## Abstract

**Simple Summary:**

Insects—as many other organisms—provide services for our societies, which are essential for our sustainable future. A classic example of an insect service is pollination, without which food production collapses. To date, though, there has often been a generalised misconception about the benefits of insects to our societies, and misunderstandings on how insects have revolutionised our cultures and thus our lives. This misunderstanding likely underpins the general avoidance, disregard for, or even deliberate attempts to exterminate insects from our daily lives. In this Perspective, we provide a different viewpoint, and highlight the key areas in which insects have changed our cultures, from culinary traditions to architecture to fashion and beyond. We then propose a general framework to help portray insects—and their benefits to our societies—under a positive light, and argue that this can help with long-term changes in people’s attitude towards insects. This change will in turn contribute to more appropriate conservation efforts aimed to protect insect biodiversity and the services it provides. Therefore, our ultimate goal in the paper is to raise awareness of the intricate and wonderful cultural relationships between people and insects that are fundamental to our long-term survival in our changing world.

**Abstract:**

Societies have benefited directly and indirectly from ecosystem services provided by insects for centuries (e.g., pollination by bees and waste recycling by beetles). The relationship between people and insect ecosystem services has evolved and influenced how societies perceive and relate to nature and with each other, for example, by shaping cultural values (‘cultural ecosystem services’). Thus, better understanding the significance of insect cultural services can change societies’ motivations underpinning conservation efforts. To date, however, we still overlook the significance of many insect cultural services in shaping our societies, which in turn likely contributes to the generalised misconceptions and misrepresentations of insects in the media such as television and the internet. To address this gap, we have reviewed an identified list of insect cultural services that influence our societies on a daily basis, including cultural services related to art, recreation, and the development of traditional belief systems. This list allowed us to formulate a multi-level framework which aims to serve as a compass to guide societies to better appreciate and potentially change the perception of insect cultural services from individual to global levels. This framework can become an important tool for gaining public support for conservation interventions targeting insects and the services that they provide. More broadly, this framework highlights the importance of considering cultural ecosystems services—for which values can be difficult to quantify in traditional terms—in shaping the relationship between people and nature.

## 1. Introduction

Ecosystem services describe the benefits that society receives from nature and include cultural services, encompassing non-material benefits including spiritual, recreational and aesthetic benefits [1]. A variety of taxa provide cultural services, amongst which—and perhaps one of the most overlooked—are insects [2]. Cultural services influence how people interact with nature and with each other. Thus, better understanding the significance of insect cultural services can help change how people perceive insects, and how future biodiversity conservation efforts can be more integrative and accommodating of insects and other invertebrates [3,4].

In general, the values given to species are linked to the experiences and familiarity of people with the species in question and with the surrounding nature into which the species is immersed [5,6]. Importantly, perceived species value, in conjunction with species charisma, often drives conservation actions [5,7] and creates inequalities in research and conservation which, for overlooked groups such as insects and other invertebrates, can be detrimental [3,8]. For instance, in Europe, funding for conservation between 1992 and 2018 through LIFE projects was limited to €150 million for invertebrates, while for vertebrates, the number was ca. 6.5×higher, amounting to €970 million [9]. Notably, funding appeared somewhat independent of species extinction risk provided that 26.3% of invertebrates are classified as critically endangered, endangered or vulnerable, in contrast to 13.3% of vertebrates [10]. Instead, species popularity remains an important predictor for conservation support in Europe [9], highlighting the cumulative effects of a species’ perception in the long-term biodiversity conservation efforts invested into it. Notwithstanding the gaps between invertebrates and vertebrates, there exist further biases *within* insects in terms of conservation attention. Conservation of insects tends to largely focus on large and colourful species while overlooking smaller and less well-known ones [3,4]. These within-insect biases are known to be damaging to native entomofauna, such as wild pollinators whose abundance can be suppressed by the high densities of honeybees [11]. Thus, conservation efforts are often associated with popularity of species, for which insects—particularly species that are smaller and less charismatic—can be largely overlooked.

Insect biodiversity conservation remains a key priority for the sustainable future of our societies given the wide range of ecosystems services insects provide [12,13,14]. Without effective conservation, we face the potential large-scale insect extinctions that will result in substantial losses in functional and phylogenetic diversity, ultimately damaging ecological networks and associated ecosystem services [15]. To prevent such extinctions, advancing technologies have been developed to allow conservationists to overcome some of the greatest challenges in approaching the conservation of insect biodiversity, including shortfalls in abundance and distribution data [16]. One such example is light detection and ranging (LiDAR) technology which quantifies the three-dimensional structures of vegetation. LiDAR has successfully been used to explain the habitat preferences of threatened butterflies in the Netherlands on a national level [17]. Other entomological applications of LiDAR include studying the relationship between forest beetle species assemblages and the environment [18]. These outputs are of great significance to understanding species distributions and evidencing conservation interventions. By utilising standardised frameworks, insights from LiDAR and other technologies can be used for the global reporting of biodiversity targets, such as the UN Sustainable Development Goals or Aichi targets [19]. Further developments are being made to automate video monitoring for the observation of plant–pollinator interactions, a technology which could aid in identifying and mitigating pollinator declines [20]. While these technologies greatly improve our ability to monitor and pinpoint trends, there is one impediment to effective insect conservation that remains pervasive: the lack of widespread regard or appreciation for insects in cultural aspects of our society. Thus, we need to integrate conservation efforts with positive public appraisal of the role and importance of insects in our societies [21]. One way in which appreciation for insects can be achieved is through changes in perception of the cultural services insects provide. For instance, while the majority of individuals and societies can recognise the significance of insect pollination to food security, relatively few truly contemplate changes in fashion, art, and other daily cultural services that are also influenced—or directly dependent upon—insects. Consequently, insects remain an unappreciated group influencing cultures. 

Insect cultural services, as with all cultural services, can be difficult to quantify in a traditional way as opposed to other ecosystem services such as pollination, which can be measured by proxies such as crop yield [22]. Economic valuations also prove suitable for some supporting services [23] and even cultural services including recreation [24]. However, many cultural services do not lend themselves to being measured by economic value or indeed any numeric metric, including cultural heritage, spirituality, and the relationships between self and the environment. More appropriate ways to quantify cultural services such as these include using indicators [25,26]. For instance, indicator frameworks have been effectively utilised on river landscapes to generate maps of cultural ecosystem services; however, this approach still does not quantify intangible elements of cultural heritage [26]. In order to encompass all areas of cultural ecosystem services, multi-disciplinary, mixed methods may be best. Mixed methods can include mixing observational studies, stakeholder meetings, and surveys [27]. Moreover, by utilising complementary methodologies, a holistic assessment of cultural services can be quantified. For example, including analysis of art and literature can allow for changes in cultural services to be examined, such as a shift toward appreciating the scenic beauty of an ecosystem within art [27]. In combination with methods such as surveys, this allows us to paint a picture of how the experiences of cultural services have changed with time in a community. Effective quantification of cultural services could better serve the need to clearly articulate the implications for environmental policies [28]. This need stems from recognising that cultural services influence how people interact with nature and with each other. To address this gap, we have firstly identified the major cultural services provided by insects and discussed their positive role in shaping societies’ functioning. To do so, we used Web of Science, Google Scholar and Scopus with the search term’s: “insects” + “cultural services” and “insects” + “culture”. Rather than an extensive literature review, our aim was to first identify papers discussing cultural services by insects, selecting examples of such services to explore in greater depth. We then discussed, with a suite of examples, the role of insect cultural services to our society, composing a perspective narrative of the field which can serve as a stepping stone for future studies aimed at quantifying their value within the broader societal context. Next, we formulated a multi-level framework that can be used to guide ways in which individuals, communities, and nations across the globe recognise and positively appraise insect cultural services within and across cultures. Overall, this paper highlights the role of insect services in shaping our societies, with the aim to influence how insects are perceived and, ultimately, help increase conservation efforts targeting insect biodiversity. 

## 2. Insect Cultural Services: How Do Insects Help Shape Us and Our Societies?

Cultural heritage and self-identity are associated with nature and cultural services, and can be a key motivator underpinning conservation efforts [29,30]. However, more often than not, insects are associated with negative emotions. Words commonly associated with flies include ‘dirty’ and ‘annoying’, while wasps are often associated with a fear of stinging [31]. Despite this disregard for insects, they play a key role in many cultural aspects of our society. These insect cultural services—associated with cultural services from other animals—are essential for human well-being. For example, increasing bird and butterfly species richness in urban green spaces is associated with self-reported psychological benefits by users of these urban areas [32], which can potentially include lowered stress, depression and anxiety [33]. Below, we provide an overview of key insect cultural services that can influence how people interact with each other and perceive nature. These services include the role of insects in shaping our traditional belief systems, our fashion trends, visual culture, media content, and our hobbies. This is of course not a comprehensive list given that insect cultural services remain an understudied field [2], but it provides a centralised narrative to demonstrate the value of insects across human cultures.

### 2.1. Insects in Traditional Beliefs and Mythology

Insects have co-existed alongside humans for millennia, and as such, we can find insects woven into traditional belief systems and mythology. One of the earliest and most well-known examples of insects in mythology comes from 8 AD, when Greek poet Ovid wrote the metamorphoses (Ov, Met, IV-VI). The key theme underpinning this collection is metamorphosis and transformation, a theme which draws clear parallels to the insect world. These works also contain insect imagery, highlighting the cultural relationship with insects. The story of Psyche—the princess of outstanding beauty—has direct links to Lepidoptera, with the Greek word Psyche translating to both ‘butterfly’ and ‘soul’. Indeed, in Greek folklore, the butterfly is representative of the soul, with the caterpillar being the body, out of which the soul emerges (Ov, Met, IV-VI). The link between the butterfly and the soul may even date further back, to 3000 BC Egypt, where butterflies were tied to the philosophy of rebirth [34]. Mythologies are important, as they allowed people to develop ideas about how insects and their place in the world came to be, providing answers to questions that seem unanswerable, such as ‘why do some insects possess unique stinging capabilities?’ In Algonquin legend, bees were afforded stingers to protect them as they laboured, while wasps pretended to be relatives of the bees in order to gain the same advantage [35]. 

These stories enrich understandings of insects and their development but also build social ties by the sharing of stories and cultures, spanning from the earliest civilisations up to present day. The prevalence of insects in mythology has also created close links between mythology and the discipline of entomology which we see in present times. Several insect species derive their name from Greek and Roman mythology. These insects include the bagworm moths (Lepidoptera: Psychidae), drawing inspiration from the aforementioned Psyche and her ties to Lepidoptera [36]. Another poignant example can be observed in Amazon ants (Hymenoptera: Formicidae), which derive their name from the all-female Amazon warriors within Greek mythology, which provides a parallel to the female-dominated ant colonies [36].

Traditional belief systems can be influenced by insects beyond the realm of mythology and storytelling. This influence is particularly evident in culinary traditions. Insects are widely consumed across the world, with at least 1681 species, from 14 orders, consumed across 102 countries [37]. For example, in Mexico, an estimated 67 species of Lepidoptera (butterflies and moths) are documented to be consumed across 29 Indigenous groups [38]. While these Lepidopterans have a higher caloric density than vertebrate food sources [39], entomophagy is more than just a nutritional practice. This entomophagy is a tradition that is learned and passed from generation to generation [40], cementing entomophagy as an important piece of cultural heritage, which has influenced the development of belief systems. 

Another instance of insects influencing traditions comes from Australian Aboriginal communities, which have used the secretion of psyllid species (known as ‘Lerps’) as a food source for generations [41]. These secretions are thought to protect psyllid nymphs from desiccation and potential predation [42,43], although lerps’ high amylose content also makes them an excellent source of energy [44]. Lerps have also shaped traditions within Aboriginal communities. Aboriginal calendars, religious, marital, and initiation ceremonies can be linked to lerp season in some instances (see [41] for review). Psyllids have different cultural roles elsewhere in the world, where they can have economic impacts as a vector of devastating plant diseases [45]. Insect pests also influence traditional beliefs such as in Nepal, where farmers have spiritual festivals that are believed to help control pest species [46]. Therefore, the relationship between people and insects shapes traditional beliefs and folklore, not only through nutrition but also with ceremonies and other activities that shape the functioning of societies. 

### 2.2. Insects in Fashion and Design

Insects have revolutionised the fashion industry. Larva of the silk moth *Bombyx mori* (Lepidoptera: Bombycidae) (aka ‘silkworm’) has been cultivated for silk since as early as the Neolithic period (3900–1700 BC) [47]. Silk production and trade originated in China but expanded to Eurasia largely due to the unique lustre, comfort, and warmth of silk [48], making it a highly sought-after fabric. Importantly, this popularity allowed not only for silk to be traded but also opened up routes for cultural exchanges [49]. The silk road itself is a key component of cultural heritage which inspires the design of silk garments up to today [50]. However, in addition to its commercial value, silk also has cultural significance and has shaped the functioning of societies. For instance, in China, upper-class officials wore silk to distinguish themselves from the lower, cotton-wearing class, while in Korea it was illegal for people of low social status to wear silk [51]. Silk also gave rise to embroidery (Figure 1a), which was further used to distinguish one’s social class within a society, where elaborate designs and colourful silk robes were reserved for those of the highest status [51]. Moreover, embroidery has been part of Chinese culture from as early as 475 BC, with examples of incredibly intricate pieces of embroidered silk having been found in tombs [52]. Together, embroidery along with the prestige and properties of silk were the seed for revolution in the fashion industry. The development of fashion was significant because fashion is a universal form of self-expression and identity, with traditions and trends that vary between communities and cultures. For instance, in India, the Navi sari has associations with national identity and socio-economic liberty [53]. Without the silk of the silkworm, we would lack the diverse fabrics that allow for such variety in self-expression such as taffeta, chiffon, charmeuse or noil. Silk is still pervasive in fashion, and not just the domain of the affluent, with 2009 being named ‘the year of the natural fibres’ [54]. Thus, we can conclude that silk has passed the test of time, with the demand for silk still increasing year upon year in the 21st century [55]. China’s economy still remains a key player, with silk exports totalling ~US$ 1 billion annually in the last decade [56]. Therefore, silk—which is a service provided by insects—has revolutionised and continues to influence our sense of identity and self-expression through fashion, thereby shaping our cultures.

Insects have prominently featured in art throughout history. Perhaps the earliest example of insects in art comes from 30,000 years ago, in the form of an etching of a cave cricket [57]. In the years since, insects have featured in several art periods with a survey of 107 museums and art databases identifying 1942 pieces of art featuring insects [58]. During the Renaissance (1400–1500), some paintings included realistic, life-sized flies on the canvas, arguably created by the artist as a Trompe-l’œil or a trick of the eye [57] and/or as symbolism for death [58].

Insects truly rose to prominence in the era of still-life paintings (1600–1800), with still-life paintings of flower arrangements featuring a number of insects (Figure 1b). In fact, of studied still-life flower books, 68–82% of the images featured insects [58]. Of these, 38% were cabbage white butterflies (Lepidoptera: Pieridae), perhaps drawing on their associations with purity. Another notable artistic period featuring insects was the Art Nouveau era (1890–1910), which was marked by a significant increase in cultural exchange between Japan and Europe, with Japanese artistic depictions of insects potentially influencing the movement in Europe [59]. The Art Nouveau period was significant in its utilisation of symbolic natural elements in art, among which insects were popular. Maurice Pillard Verneuil was commendable for a taxonomically diverse use of insects in his Art Nouveau work (Figure 1c), with cicadas, beetles, butterflies, and dragonflies featuring in works [59]. This period also saw a surge in insect-themed three-dimensional art, including vases, bowls, and furniture [59].

Given that 11 insect orders engineer structures and nests from natural materials [60], it is no surprise that insects can serve as inspiration to human engineers and architects, especially in the ongoing sustainability movement. Using biological entities for design is known as biomimetics and social insects are particularly interesting to engineers in this area because they have solved key functional challenges associated with sociality during evolution [61]. For instance, honeycomb structures have long been studied by humans because of their lightweight, porous structures that maximises space in an energy efficient manner and in recent decades, has increasingly been incorporated into human-made designs [62]. This inspiration is exemplified in architecture, with honeycomb structures used to design buildings that are multi-functional, energy efficient and importantly, sustainable [63]. These properties are possible owing to the qualities of the honeycomb, which allows for lightweight designs with enhanced thermal insulation and energy absorption [62]. Furthermore, insects also provide more artistic inspirations for infrastructure. For instance, the design of the Tiszavirag bridge in Hungary is claimed to mimic the shape of the mayfly *Palengenia longicauda* (Ephemeroptera: Palingeniidae), the largest mayfly species in Europe and one that spawns annually on the Tisza river (Figure 1d) [64]. Hundreds of people cross the bridge daily and more importantly, people also gather on the bridge every year to appreciate *P. longicauda* mass spawning (known as ‘Tisza blooming’). In fact, this event (along with the mayfly species) has been deemed a unique Hungarian national value [65]. The Tisza blooming is also present in art, such as folk music, where the mayfly poses as a metaphor for the brevity of life and love [66]. *P. longicauda* is therefore an example of an insect that has inspired cultural artifacts simply by its unique ecology and life history. Currently though, *P. longicauda* has been lost from 98% of its range, existing only in a few isolated populations [67], putting at risk the long-term sustainability of the Tisza blooming and, more broadly, the provisioning of such an important cultural service.

### 2.3. Insects in Media

In popular culture, insects can be given an unsavoury reputation, such as in ‘insect fear films’ which capitalise upon hyperbolic fears of insects and their morphology [68]. These films contribute to the fear of insects and also influence the horror movie industry. For instance, the insect fear film festival has been an annual event since 1984, attracting thousands of horror movie and insect fans [69]. The festival poses a unique opportunity to both enjoy insect fear films and dispel (or even reinforce) some of the misconceptions about insects, fostering a greater appreciation (or depreciation) for insects. Other movies genres have also drawn inspiration from insects including documentary films. These are less abundant than insect fear films, possibly because they do not capture the public’s imagination and attention in the same manner as the horror genre [68]. Anthropomorphised insects capture audiences’ attention, particularly in animated films. For instance, bees have been positively (albeit somewhat inaccurately) portrayed in the Bee Movie, thereby stimulating further interest in the ecology of bees [70]. Other elements of media have been inspired by insect morphology, including comic books, which have drawn inspiration from insects for characters in roles of superheroes and/or villains in equal measure [71]. This inspiration emerges from the unique abilities of insects including flight, stinging, secretion production, and metamorphosis. For example, ant-man has increased strength with a small body mass, and the ability to communicate and cooperate with others, helping reinforce positive stereotypes of his namesake insect. Insects and their attributes are also a feature of music including in artist names, albums and song titles from rock and roll [72]. Music, particularly metal and punk, plays upon ideas of fear and revulsion but also references insects for their beauty and other positive qualities [72]. Music has less visual impact than cinema, arguably reducing the effectiveness of frightening imagery. This medium allows the audience to experience a variety of insect-inspired metaphors—both positive and negative—without the visual triggers provided by movies and other forms of visual arts.

Lastly, storytelling is a key tool to impart knowledge, tradition, and moral values upon children, which is especially apparent within Indigenous culture. In Nepal, stories such as ‘the dung beetle and the cicada’ are used to highlight morality and work ethic [46]. Similarly, the study of ant tracks poses an opportunity to teach patience and endurance [73]. Stories and other cultural services that portray insects in a positive light are important, as they allow for children to relate with and engage with the surrounding entomofauna. This engagement can also be achieved by story-telling in modern literature. Many published children’s books involving insects focus upon the teaching of a particular life lesson [74]. Interestingly, the fear factor of insects has also invaded literature, with a great number of science fiction and fantasy novels drawing upon the same hyperbolic fears as the horror film genre [74]. Overall, these insect-inspired media are significant, because they provide an accessible opportunity for children and adults to learn about and engage with insects.

### 2.4. Insects in Recreation and Hobbies

The value of insects can also be imparted through recreational activities and hobbies. For instance, several games in Indigenous cultures involve the collection of insects [46,73,75]. Additionally, ‘pond dipping’ for freshwater invertebrates is a recreational activity in many places, involving the identification of pond invertebrates in pond water, allowing for contact with insects and education of freshwater ecosystems [76]. These activities are important for long-term conservation of insect biodiversity because contact with insects can reduce (or perhaps increase) their fear and perceived danger [77,78]. Another recreational activity that is important for the long-term conservation of insects is participation in citizen science projects. People of all cultures have long enjoyed collecting insect specimens, giving entomological collections an ever-expanding number of applications [79]. Butterflies are an especially common focal point for citizen science projects, such as the eButterfly project in North America, which has collated more than 400,000 observations from over 5000 citizen scientists [80]. These citizen science projects promote recreational benefits, but also provide opportunities for democratising scientific knowledge and strengthening social capital [81].

Fishing is a common recreational (and professional) activity across the world. In England and Wales, it is estimated that game angler’s expenditure totals over £400 million, making it a sport of considerable economic importance [82]. It is also a sport which has been developed with influence from freshwater insects. Insects have served as models for fly fishing lures [83] and many angling events occur in synchrony with the annual emergence of mayflies or stoneflies [83]. Insects have therefore influenced and—to some extent, inspired the invention of—fly fishing. It is important to mention that many anglers consider fly fishing to be a spiritual activity, referring to nature as something sacred to them [84]. Fly fishing can therefore be described as recreational but also personal identity activity, and has been so for at least two thousand years [85] with demonstrated healing benefits with PTSD patients and other traumas [86,87].

## 3. Bringing Insects Back into Culture: A Multi-Level Framework

Overcoming the general public’s disregard for insects is a significant obstacle to insect biodiversity conservation [16]. Nonetheless, it is possible to foster activities and materials that adopt a more positive and constructive view of insects and their roles in the interaction between people and nature. Above, we provided a description of the many ways in which insects influence our culture. In this section, we formulate a multi-level framework (Figure 2) that can be used as a guide to promote insect cultural services and, in the long term, and contribute to effective conservation policies and efforts to protect insect biodiversity.


(a)Individual-level actions


Our perceptions of insects are shaped through experience. Having negative experiences with insects can create a dislike for insects [88], which in turn reduces interest in learning about insects [89]. However, positive experiences can greatly reduce the perceived danger of insects [78,90] and create positive attitudes towards their conservation [77,78]. Therefore, to dispel negative stereotypes about insects and maximise positive perceptions, we should recognise and celebrate the positive cultural contributions of insects. Media that portrays insects in a positive light can be powerful in driving perceptions, including books and films that draw inspiration from the positive characteristics of insects, such as cooperation. Another example is by insect-friendly gardening practices, which provide habitat for insects, contribute to sense of place (e.g., well-being, relationship with nature) and give recreational enjoyment [91].


(b)Community level actions


Cultural services contribute to values such as sense of place and heritage, which define and give meaning to communities worldwide. As a result, actions and campaigns that can link insects to the cultural heritage of a community are likely to motivate people to engage in environmental management activities that are beneficial to insects [92]. Communities can be made aware of the wide range of cultural services by a variety of means. One such example is the usage of story maps to communicate the cultural value of coastlines [27]. Approaches like this reiterate the value of species and habitats to communities and can put that value in the context of conservation. Story maps could be an especially effective tool to spread awareness of the many elements of culture that have been shaped (and continue to be shaped) by insects. This level builds upon the first level of the framework, whereby individuals who have developed a positive perception about insects and their value are also more likely to share recreational, educational or spiritual activities within their communities.


(c)National-level actions


The national level is where many policy decisions that will address conservation are made. The cultural services provided to communities by insect biodiversity could be quantified when making decisions for conservation. Recognising cultural services would allow decision makers to determine whether particular interventions will enhance or detract from cultural services. Cultural services of importance and value to the community can be identified by engaging in structured discussions [93] or using discourse-based methods [94]. A wide range of community members should be involved such as the disabled or homeless within a community, as they may have differing perceptions of cultural services [95,96]. Widespread community engagement will allow decision makers to act accordingly such as considering the accessibility of pollinator-friendly allotments within the community. By quantifying cultural services and identifying their importance to the community, conservation can be placed in the context of benefitting both society and insects and strengthening the cultural ties between them.


(d)Global-level actions


The prior three levels culminate on the global level. Building positive relationships between humans and insects and motivating stakeholders to engage in the conservation of insects across nations will support large-scale action for conserving insect biodiversity. By recognising cultural services and valuing insects, we can conserve the range of ecosystem services provided by insects, which in turn will aid in aspirations of sustainability. These aspirations include the achievement of the UN Sustainable Development Goals, for which insects are of critical importance [13]. Appreciating the cultural impact of insects and conserving their numbers will also conserve supporting services such as pollination. Conservation of these services will underpin SDGs including #1 End Poverty and #2 End Hunger, exemplifying the importance of appreciating insects, an appreciation which can be fostered by recognising the contributions of insects to culture.

## 4. Conclusions

In this paper, we discussed several examples of insect cultural services and developed a multi-level framework for increasing awareness and interest in insect conservation via cultural services. It is important that policy makers (e.g., parliaments and associated stakeholders) recognise and integrate insect cultural services when designing environmental policies for conserving biodiversity. Integration of cultural services could involve utilising emerging strategies and frameworks for recognising or valuing cultural services in decision-making processes [28,97]. Uptake of these approaches has the potential to reduce biases within conservation that work to the detriment of insects [6,8,9], creating a society that recognises the value of insects beyond economic value. Currently, the UK National Ecosystem Assessment has explored methodologies for incorporating cultural services into decision-making processes, which include participatory mapping, and cultural indicators for measuring the contribution of cultural services to communities [98]. More studies are needed in this area, both to identify and quantify insect cultural services, and to best integrate their value into policies.

## Figures and Tables

**Figure 1 insects-12-00377-f001:**
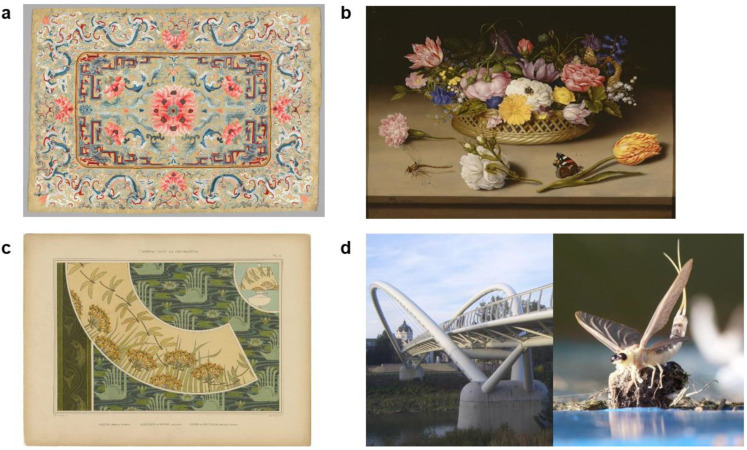
Cultural services provided by insects in the visual arts: (**a**) 19th-century Chinese embroidered silk (Cleveland Museum of Art: https://www.jstor.org/stable/10.2307/community.24604198 (accessed on 9 April 2021)); (**b**) 17th-century floral still life by Ambrosius Bosschaert the Elder (Dutch, 1573–1621) featuring several insect species (J. Paul Getty Museum: https://www.jstor.org/stable/10.2307/community.15987452 (accessed on 9 April 2021)); (**c**) Plate from L’animal dans la décoration by Maurice Pillard Verneuil (French illustrator, 1869–1942) from the Art Nouveau period (https://www.jstor.org/stable/10.2307/community.29393027 (accessed on 9 April 2021)); (**d**) Tiszavirag bridge in Hungary (www.wikidata.org (accessed on 9 April 2021)) inspired by the *Palengenia longicauda* annual mass spawning (right) (www.wikipedia.org (accessed on 9 April 2021)).

**Figure 2 insects-12-00377-f002:**
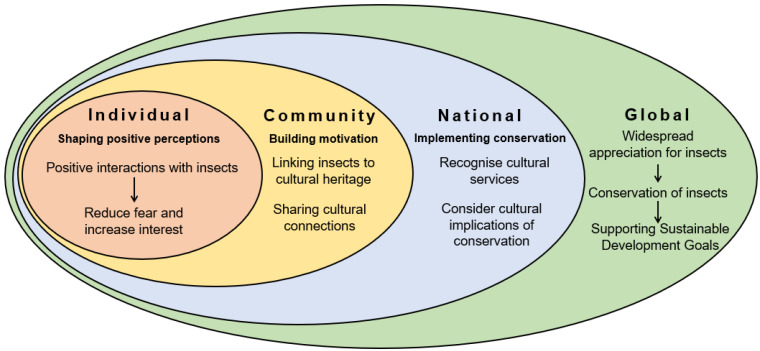
Multi-Level Framework: On the individual level, positive perceptions of insects can be fostered by positive experiences with insects. On the community level, experiences of cultural services can influence motives to conserve nature. On the national level, conservation for insects should be designed with the motivations of stakeholders and cultural services in mind. Globally, this framework will allow for effective conservation of insects.

## Data Availability

No new data were created or analysed in this study. Data sharing is not applicable to this article.

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
