# Peer review of "Insect Cultural Services: How Insects Have Changed Our Lives and How Can We Do Better for Them"

_insects, 2021, doi:10.3390/insects12050377_

Round 1

Reviewer 1 Report

In general terms, it is a fascinating manuscript. It is well written, and the figures are valuable. However, some important things are correct and precise to improve it and make possible its acceptance.

Major:

L80-87: It is necessary to specify and detail the methodology used for the bibliographic search of insects' cultural services. What were the search engines consulted, the keywords, the method of selection, and refinement of the articles found?

Figure 1 is too small; make it bigger and separate from the text. The text in the figure must be in a bigger font.

The multi-level framework that you build could be interesting to add a global level related to the world-conservation of insects and a higher awareness of the ecosystem services that insects provide.

Minor:

L4: There is a missing * in the affiliation. Is this related to the corresponding author?

L23: e.g., must be in italic.

L39: add “ecosystem services” and “societies”

L48: add Schowalter et al. 2018. Basic and Applied Ecology, 26: 1-7

L70: you miss the citation of reference 10

L74: add Cardoso et al. 2020 Biological Conservation and Samways et al. 2020 Biological Conservation and discuss the implications of a potential insect extinction.

L95: change to “a key role in many cultural”

L114: change to “This entomophagy”

L121: change to “marital, and”

L122: change to “(see 23 for review).”

L130: add the family of the moth.

L133: change to “comfort, and”

L143: separate 475 BC

L152: you miss the citation of reference 35

L157: add the family of the mayfly.

L233: e.g., must be in italic.

Reviewer 2 Report

The manuscript entitled “Insect cultural services: how insects have changed our lives and  how can we do better for them” is reviewed here. overall, this paper is more suitable after carry out the major changes for a journal of Insects.

  1. Introduction section needs to develop recent updates with the insect cultural services as well as advanced technologies using to conserve the biodiversity.
  2. Add sub heading of Insect sustainable agriculture management
  3. Even if the subject can be itself interesting, the MS has serious flaws, both in writing and the section arrangements and it is looks like a prospective rather than original manuscript.
  4. What is the different strategy of insect cultural services available?
  5. What is the policy-makers in insect cultural services?
  6. Over all the manuscript needs to edit the mentioned changes and check the language corrections through native speaker.

Reviewer 3 Report

Significantly more depth and breadth should to be provided about cultural impact of insects. This is true in all the categories of cultural services. Although this is stated as a “Perspective” publication, it lacks any type of survey or metrics which would serve to validate the points the authors are trying to make.

  • Traditional beliefs and culinary. This is a somewhat strange pairing. There is no mention of the role that insects play in many traditional societies, or that insects have played in Greek and Roman mythology. The entire process of metamorphosis and Psyche in belief systems, as well as literature and visual art should be mentioned. For culinary applications there are a few examples, but it would be much more powerful if a survey was done of the number of countries and/or cultures where insects are part of the cuisine. It also would be important to mention that companies are looking towards insects as an alternative source of high quality and low cost protein.
  • Fashion and architectural design. While the discussion of silk is good, this is a very small part of the art world that insects have influenced. There are some notable periods in art (e.g. Art Nouveau) where insects featured prominently. In other cases, e.g. still life paintings in the Renaissance, insects were included for symbolic reasons. For architecture, while the mention of the mayfly bridge is valid and interesting, the paper completely ignores the influence of insect architecture (e.g. the honeycomb design) on human architecture and design.
  • Insects in media. There is no mention of music, which is a major oversight. The Insect Fear Festival is notable because it plays on the fear of insects, but how do “fear” films stack up vs. other types of films including insects?
  • Conservation and hobbies. Again, there is superficial and scant consideration of this topic. What about the many Citizen Science projects involving insects, Lost Ladybug Project, Giant Sunflower Project, the Firefly Festival, INaturalist projects.

The framework of Individual, Community and National level actions is good, but the examples given are vague and not particularly convincing. For example, books and films about the positive characteristics of insects. These already exist. The challenge is (at the individual, community and national level) how do you make these voices heard above those that derive profit from keeping people afraid of insects?

The authors mention the difficulty in quantifying cultural services vs. ecosystem services. I am not convinced that cultural “services” is the best way to frame the impact of insects on humans. Perhaps a better way is to conduct a robust study of the cultural realm (vs. the superficial study presented here) to form a complete picture of what insects have contributed to the development of civilizations, and what they have taught us about ourselves.

Round 2

Reviewer 2 Report

The authors did a good job to revise the manuscript 

Reviewer 3 Report

The authors have taken the reviewers' comments under consideration and made a significant number of improvements.

Author Response

Thank you. Please see attached response to the Editors' comments.
